# LOTTERY TICKETS CAN HAVE STRUCTURAL SPARSITY

## ABSTRACT

The lottery ticket hypothesis (LTH) has shown that dense models contain highly sparse subnetworks (i.e., *winning tickets*) that can be trained in isolation to match full accuracy. Despite many exciting efforts being made, there is one "common-sense" seldomly challenged: a winning ticket is found by iterative magnitude pruning (IMP) and hence the resultant pruned subnetworks have only unstructured sparsity. That gap limits the appeal of winning tickets in practice, since the highly irregular sparse patterns are challenging to accelerate on hardware. Meanwhile, directly substituting structured pruning for unstructured pruning in IMP damages performance more severely and is usually unable to locate winning tickets.

In this paper, we demonstrate **the first positive result** that a structurally sparse winning ticket can be effectively found in general. The core idea is to append "post-processing techniques" after each round of (unstructured) IMP, to enforce the formation of structural sparsity. Specifically, we first "re-fill" pruned elements back in some channels deemed to be important, and then "re-group" non-zero elements to create flexible group-wise structural patterns. Both our identified channel- and group-wise structural subnetworks win the lottery, with substantial inference speedups readily supported by practical hardware. Extensive experiments, conducted on diverse datasets across multiple network backbones, consistently validate our proposal, showing that **the hardware acceleration roadblock of LTH is now removed**. Specifically, the structural winning tickets obtain up to $\{64.93\%, 64.84\%, 64.84\%\}$ running time savings at $\{36\% \sim 80\%, 74\%, 58\%\}$ sparsity on {CIFAR, Tiny-ImageNet, ImageNet}, while maintaining comparable accuracy. All the codes and pre-trained models will be publicly released.

## 1 INTRODUCTION

Recently, the machine learning research community has devoted considerable efforts and financial outlay to scaling deep neural networks (DNNs) to enormous sizes (175 billion parameter-counts in GPT-3 (Brown et al., 2020)). Although such overparameterization simplifies the training of DNNs and dramatically improves their generalization (Bartlett et al., 2021; Du et al., 2018; Kaplan et al., 2020), it may severely obstruct the practical usage on resource-limited platforms like mobile devices, due to its large memory footprint and inference time (Hoefler et al., 2021). Pruning as one of the effective remedies can be dated back to LeCun et al. (1990): it can eliminate substantial redundant model parameters and boost the computational and storage efficiency of DNNs.

Such benefits drive numerous interests in designing model pruning algorithms (Han et al., 2015a;b; Ren et al., 2018; He et al., 2017; Liu et al., 2017). Among this huge family, an emerging representative studies the prospect of training *sparse subnetworks* in lieu of the full dense models without impacting performance (Frankle & Carbin, 2019; Chen et al., 2020b). For instance, Frankle & Carbin (2019) demonstrates that dense models contain sparse, matching subnetworks (Frankle et al., 2020a) (a.k.a. *winning tickets*) capable of training in isolation from the original initialization to match or even surpass the full accuracy. This phenomenon is referred to as the *lottery tickets hypothesis* (LTH), which indicates several impressive observations: ($i$) usually extreme sparsity levels (e.g., $90\%$, $95\%$) can be achieved without sacrificing the test accuracy; ($ii$) the located winning ticket maintains undamaged expressive power as its dense counterpart, and can be easily trained from scratch or early-epoch weights (Renda et al., 2020; Frankle et al., 2020a) to recover the full performance. These advances are positive signs about the substantial potential of sparse DNNs.

However, almost all LTH literature investigates unstructured sparsity only. In practical scenarios, it brings little hardware efficiency benefits due to the poor data locality and low parallelism (He et al., 2017; Mao et al., 2017; Wen et al., 2016) caused by highly irregular sparse patterns. Meanwhile, most of the accelerators are optimized for dense matrix operations (Han et al., 2016), which means there is limited speedup for unstructured pruned subnetworks even the sparsity level exceeds 95% (Wen et al., 2016). Structural pruning (He et al., 2017; Liu et al., 2017) as an alternative to exploring sparse subnetworks, removes the entire filter or channel in DNNs to gain more computational efficiency at the cost of (more) accuracy degradation. As shown in Figure 1, traditional channel-wise structural pruning approaches (He et al., 2017; Liu et al., 2017; Bartoldson et al., 2019; Molchanov et al., 2019) quickly degrade performance and cannot lead to winning tickets, which was also echoed in You et al. (2020).

In our paper, we present the first study into the *structural lottery tickets*, which explores hardware-friendly structural sparsity (including channel-wise and group-wise patterns) in order to find lottery tickets. Specifically, we start from unstructured sparse subnetworks, and then adopt proposed *refilling* techniques to create channel-wise structural sparsity by growing back the pruned elements within the most important channels and abandoning the rest. Our results (Section 4) show such refined channel-wise structural subnetworks win the lottery at a moderate sparsity level with $\sim 50\%$ running time savings on an Nvidia 2080 TI. In order to push the compression ratio higher, we introduce a *regrouping* algorithm based on hypergraph partitioning (Rumi et al., 2020) to establish group-wise structural patterns which are more amenable to pruning due to the shape flexibility of grouped dense blocks. These group-wise structural winning tickets achieve $\sim 60\%$ running time savings

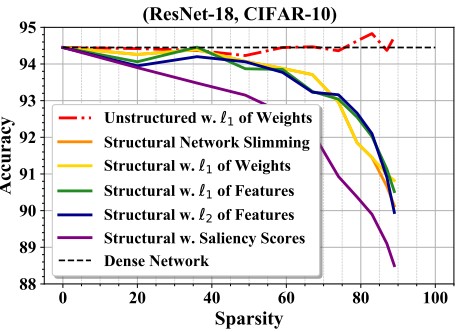

Figure 1: Achieved test accuracy over different sparsity levels of diverse unstructured and structural subnetworks. Sparse models from classical channel-wise structural pruning algorithms (He et al., 2017; Liu et al., 2017; Bartoldson et al., 2019; Molchanov et al., 2019) can not match the full accuracy of the dense model (dash line).

at $50\% \sim 80\%$ sparsity without any performance degradation compared to the dense models. Our main contributions lie in the following aspects:

- To our best knowledge, we are the first to demonstrate the existence of structurally sparse winning tickets at non-trivial sparsity levels (i.e., $> 30\%$), and with both channel-wise and group-wise sparse patterns.
- We propose the *refilling* technique and introduce the *regrouping* algorithm to form channel-wise and group-wise structural sparsity, respectively. Such refined structural subnetworks match the trainability and expressiveness of dense networks, while enabling the inference speedup on practical hardware platforms like GPU machines.
- Extensive experiments validate our proposal on diverse datasets (i.e., CIFAR-10/100, Tiny-ImageNet, and ImageNet) across multiple network architectures, including ResNets, VGG, and MobileNet. Specifically, our structural winning tickets achieve $53.75\% \sim 64.93\%$ GPU running time savings at $45\% \sim 80\%$ channel- and group-wise sparsity.

## 2 RELATED WORK

**Pruning.** Network pruning is a technique that aims at eliminating the unnecessary model parameters (Blalock et al., 2020), which can effectively shrink models for the deployment on resource-constrained devices (LeCun et al., 1990; Hanson & Pratt, 1988). Pruning algorithms are roughly categorized into two groups: (1) unstructured pruning (LeCun et al., 1990; Han et al., 2015a;b; Ren et al., 2018; Zhang et al., 2018) with irregular sparse patterns; (2) structural pruning (He et al., 2017; Liu et al., 2017; Li et al., 2016; Hu et al., 2016; Wen et al., 2016; Hong et al., 2018) with structural sparse patterns such as channel-wise, block-wise, column-wise, etc..

Within the group of unstructured pruning, Han et al. (2015a;b) remove insignificant connections of models in the post-training stage, with respect to certain heuristics like weight/gradient magnitudes; during training sparsification is also another popular trend for pruning by leveraging $\ell_0$ regularization (Louizos et al., 2017) or alternating direction method of multipliers (ADMM) (Ren et al.,

2018; Zhang et al., 2018). Recently, several pruning-at-initialization methods (Wang et al., 2020; Lee et al., 2019b; Tanaka et al., 2020) are proposed to identify critical unstructured connections for gradient-flow preserving, without any training. Although the unstructured sparse model has superior performance, it usually suffers from poor data locality and low parallelism (He et al., 2017; Mao et al., 2017; Wen et al., 2016), which make it hard to speed up in real-world applications.

On the contrary, structural pruning is more hardware-friendly at the cost of notable accuracy loss when the compression ratio increases. He et al. (2017); Liu et al. (2017) slim the network channels via $\ell_1$ regularization, and Bartoldson et al. (2019) selects important channels according to heuristics of feature maps. To combine the benefits of structural and unstructured pruning, hybrid pruning strategies have been introduced to pursue more general structural spares patterns which are also capable of acceleration. For example, convolution kernels with half regular sparsity (Chen et al., 2018) or pattern-based structural sparsity (Ma et al., 2020) or vector-wise (Zhu et al., 2019) and group-wise (Rumi et al., 2020) regular sparsity.

**The lottery tickets hypothesis (LTH).**   The lottery ticket hypothesis (LTH) (Frankle & Carbin, 2019) conjectures that there exists a sparse subnetwork called winning ticket within a dense network, whose performance can match with the dense network when training from the same initialization. With the assistance of weight rewinding techniques (Renda et al., 2020; Frankle et al., 2020a), the original LTH can be scaled up to larger networks and datasets. The existence of winning tickets are broadly verified under diverse contexts, such as image classification (Frankle & Carbin, 2019; Liu et al., 2019; Wang et al., 2020; Evci et al., 2019; Frankle et al., 2020b; Savarese et al., 2020; You et al., 2020; Ma et al., 2021a; Chen et al., 2020a), object detection Girish et al. (2020), natural language processing Gale et al. (2019); Yu et al. (2020); Prasanna et al. (2020); Chen et al. (2020b;c), generative adversarial networks Chen et al. (2021d); Kalibhat et al. (2020); Chen et al. (2021a), graph neural networks Chen et al. (2021b), reinforcement learning Yu et al. (2020), and life-long learning Chen et al. (2021c). However, all of the above LTH literature only locate *unstructured* sparse winning tickets, which can hardly bring hardware efficiency boost on real-world applications.

As the most related work, You et al. (2020) finds structural winning tickets at only low sparsity levels around $30\%$ in few cases. It again reveals the complication and difficulty of identifying computation-friendly sparse patterns. Another concurrent work (Alabdulmohsin et al., 2021) investigates a generalized LTH with weight space factorization, which is orthogonal to our work.

**Sparse convolutional neural network (CNN) acceleration on GPU.**   Previous works have explored the acceleration of sparse convolution operations in two different directions. One direction is to design efficient implementation of unstructured pruned networks for improved data locality and utilization of hardware (Chen, 2018; Park et al., 2016). For example, Dong et al. (2019) proposes "Acorns" to accelerate the sparse computations of convolution kernels with an input sparsity. Peng et al. (2017) has proposed a matrix splitting algorithm for efficient CNN inference. Nvidia's cuSPARSE[1] library contains various efficient sparse matrix computation algorithms like SpMM on GPUs, drawing great attention in efficient scientific computing. Furthermore, advanced approaches are developed based on SpMM, such as Adaptive Sparse Tiling (ASpT) (Hong et al., 2019). ASpT significantly improves the data usage of SpMM and achieves the current state-of-the-art performance among SpMM implementation variants. Another direction focuses on more hardware-friendly pruning methods (Chen et al., 2018; Ma et al., 2020; Niu et al., 2020). During the model pruning, these works aim to maintain certain regular sparse patterns, which benefit the hardware processing/computing of corresponding sparse matrices. However, Chen et al. (2018) achieves unsatisfactory compression ratio, while the pruning methods used in Ma et al. (2020) and Niu et al. (2020) require dedicated compiler optimization to accelerate network execution.

## 3   METHODOLOGY

### 3.1   NOTATIONS AND PRELIMINARIES

**Sparse subnetworks and pruning methods.**   In this paper, we mainly follow the routine notations in Frankle & Carbin (2019); Renda et al. (2020). For a network $f(x; \theta)$ with input samples $x$ and model parameters $\theta$, a sparse subnetwork is a network $f(x; m \odot \theta)$ with a binary pruning mask $m \in \{0, 1\}^{|\theta|}$, where $\odot$ is the element-wise product. In other words, it is a copy of dense network

---

[1]https://docs.nvidia.com/cuda/archive/10.2/cusparse/index.html

$f(x; \theta)$ with some weights fixed to 0. If the non-fixed remaining weights are distributed irregularly, we call it **unstructured** sparse patterns (e.g., Figure 2, *left*); if they are clustered into channels or groups, we name it **structural** sparse patterns (e.g., Figure 2, *right*).

To obtain the desired sparse subnetworks, we consider and benchmark multiple classical pruning algorithms: (1) *random pruning* (RP) which usually works as a necessary baseline for the sanctity check (Frankle & Carbin, 2019); (2) *one-shot magnitude pruning* (OMP) by eliminating a part of model parameters with the globally smallest magnitudes (Han et al., 2015a); (3) *the lottery ticket hypothesis* (Frankle & Carbin, 2019) with iterative weight magnitude pruning (LTH-IMP or IMP for simplicity) (Han et al., 2015a). As adopted in LTH literature (Frankle & Carbin, 2019), we identify the sparse lottery tickets by iteratively removing the 20% of remaining weight with the globally smallest magnitudes, and rewinding model weights to the original random initialization (Frankle & Carbin, 2019) or early training epochs (Frankle et al., 2020b; Chen et al., 2020a). In this paper, the model weights are rewound to the eighth epoch (i.e., the 5% of the entire training process) for all CIFAR, Tiny-ImageNet, and ImageNet experiments. (4) *pruning at initialization* mechanisms. We choose several representative approaches such as SNIP (Lee et al., 2019a), GraSP (Wang et al., 2020), and SynFlow (Tanaka et al., 2020), which explore sparse patterns at random initialization with some gradient flow based criterion. (5) *Alternating Direction Method of Multipliers* (ADMM) for punning. It is a well-known optimization-based pruning method (Niu et al., 2020; Zhang et al., 2018), which can obtain superior compression ratios with little performance degradation for deep neural networks. Note that all pruning approaches are mainly conducted over networks without counting their classification heads (Frankle & Carbin, 2019; Ma et al., 2021b).

**Structural winning tickets.** We begin by extending the original lottery tickets hypothesis to the context of structural sparse patterns. A subnetwork $f(x; m \odot \theta)$ is a structural winning ticket for an algorithm $\mathcal{A}_t^{\mathcal{T}}$ if it satisfies: ① training subnetworks $f(x; m \odot \theta)$ with algorithm $\mathcal{A}_t^{\mathcal{T}}$ results in performance measurement on task $\mathcal{T}$ no lower than training dense networks $f(x; \theta)$ with algorithm $\mathcal{A}_t^{\mathcal{T}}$, where $\theta$ is the original random initialization $\theta_0$ or early rewound weights like $\theta_{5\%}$, and $t$ is the training iterations; ② the non-zero elements in pruning mask $m$ are clustered as channels, groups or other hardware-friendly structural patterns.

**Implementation details.** We conduct experiments on diverse combinations of network architectures and datasets. Specifically, we adopt Wide-ResNet-32-2 (Zagoruyko & Komodakis, 2016) (or WRN-32-2), ResNet-18 (He et al., 2016) (or RN-18), MobileNet-v1 (or MBNet-v1) (Howard et al., 2017), and VGG-16 (Simonyan & Zisserman, 2014) on both CIFAR-10 (Krizhevsky et al., 2009) and CIFAR-100 datasets. ResNet-50 (or RN-50) is evaluated on both Tiny-ImageNet (Le & Yang, 2015) and ImageNet (Deng et al., 2009) datasets. Table 1 includes more training and evaluation details of our experiments.

Table 1: Implementation details which follow the standard settings in Ma et al. (2021b).

| Settings | CIFAR-10 | | | | CIFAR-100 | | | | Tiny-ImageNet | ImageNet |
|---|---|---|---|---|---|---|---|---|---|---|
| | WRN-32-2 | RN-18 | MBNet-v1 | VGG-16 | WRN-32-2 | RN-18 | MBNet-v1 | VGG-16 | RN-50 | RN-50 |
| Batch Size | 128 | 128 | 128 | 128 | - | - | 64 | - | 32 | - |
| Weight Decay | $1 \times 10^{-4}$ | $1 \times 10^{-4}$ | $1 \times 10^{-4}$ | $2 \times 10^{-4}$ | $2 \times 10^{-4}$ | $2 \times 10^{-4}$ | $2 \times 10^{-4}$ | $5 \times 10^{-4}$ | $5 \times 10^{-4}$ | $1 \times 10^{-4}$ |
| Learning Rate | $0.1; \times 0.1$ at 80,120 epoch of total 160 epochs | | | | | | | | | |
| Optimizer | SGD (Ruder, 2016) with a momentum of 0.9 | | | | | | | | | |
| Model Size | 1.86 M | 11.22 M | 3.21 M | 14.72 M | 1.86 M | 11.22 M | 3.21 M | 14.72 M | 25.56 M | 25.56 M |

### 3.2 REFILLING FOR STRUCTURAL PATTERNS

It is well-known that the irregular sparsity patterns from unstructured magnitude pruning block the acceleration on practical hardware devices. To overcome the limitation, we propose a simple *refilling* strategy to reorganize the unstructured sparse patterns and to make them more hardware friendly. Specifically, we first select important channels from the unstructured subnetwork according to certain criteria. The number of picked channels are depended on the desired sparsity level. Then, the pruned elements are grown back to be trainable (i.e., unpruned) and are reset to the same random initialization or early rewound weights. Lastly, the rest parameters in the remaining insignificant channels will be removed. In this way, we refill important channels and empty the rest to create a channel-wise structural sparse pattern that essentially brings computational reductions. Note that the picking criterion can be the number of remaining weights in the channel, or the channel's weight statistics or feature statistics or salience scores, which are comprehensively investigated in the ablation (Section A2). The complete pipeline and illustration are summarized in Algorithm 2 and Figure 2, respectively.

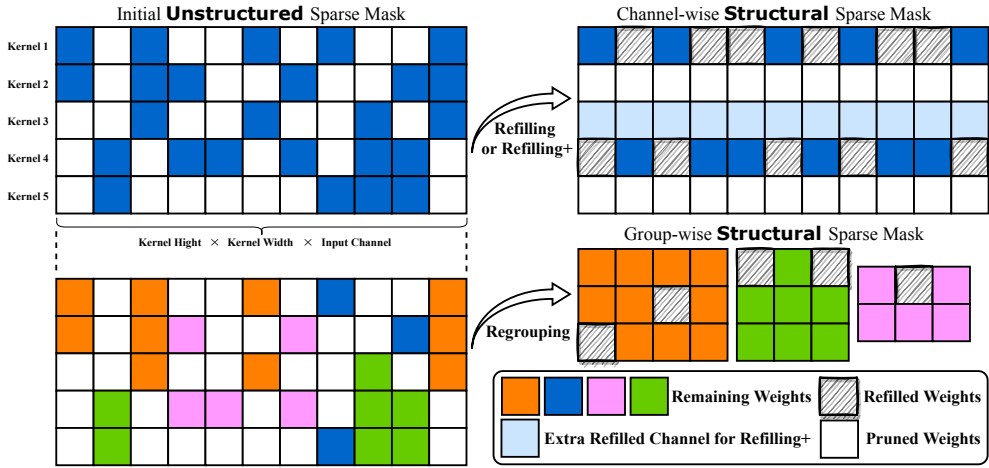

Figure 2: Overview of our proposals including refilling, refilling+, and regrouping, which turn unstructured sparse mask into channel-wise and group-wise structured sparse masks.

---

**Algorithm 1:** `IMP` with rewinding step $i$

**Input:** $f(x; \theta_0)$, unstructured sparsity $s$
**Output:** $f(x; m \odot \theta_i)$

1 Set the pruning mask $m = \mathbf{1} \in \mathbb{R}^{|\theta|}$
2 Train $f(x; \theta_0)$ to rewinding step $i$:
  $f(x; \theta_i) = \mathcal{A}_i^{\mathcal{T}}(f(x; \theta_0))$
3 **while** *not reach sparsity $s$* **do**
4   Train $f(x; m \odot \theta_i)$ to step $t$:
    $f(x; m \odot \theta_t) = \mathcal{A}_{t-i}^{\mathcal{T}}(f(x; m \odot \theta_i))$
5   Pruning 20% of remaining weight of
    $m \odot \theta_t$, and update $m$
6 **end**

---

**Algorithm 2:** `IMP-Refill(+)`

**Input:** $f(x; m \odot \theta_i)$ with unstructured
  sparsity $s$ from Algorithm 1
**Output:** $f(x; m \odot \theta_i)$ with channel-wise
  structural mask $m$ at sparsity $\tilde{s}$

1 Calculate importance scores of each
  channel according to certain criterion
2 Pick top-$k$ channels in $m$, refill back their
  0 (pruned) elements with 1 (trainable)
  and update $m$, maintaining $\tilde{s} \sim s$
3 Pick and refill back extra channels in $m$
  with $\tilde{s}^+ < s$ # Optional for `Refill+`

---

**Algorithm 3:** `IMP-Regroup`

**Input:** $f(x; m \odot \theta_i)$ with unstructured
  sparsity $s$ from Algorithm 1,
  hyperparameters $t_1$, $t_2$, $b_1$, and $b_2$
**Output:** $f(x; m) \odot \theta_i$ with group-wise
  structural mask $m$ at sparsity $s^*$

1 **while** *dense block can be found* **do**
2   Divide the rows of the sparse pruning
    mask $m$ into $t_1$ groups using hypergraph
    partitioning (hMETIS)[a]
3   **for** *group $c_i \in \{c_1, c_2, \ldots, c_{t_1}\}$* **do**
4     **if** *$c_i$ has $\geq b_1$ rows* **then**
5       Select columns in $c_i$ that has no
        less than $t_2$ non-zero items
6       **if** *$\geq b_2$ columns are selected* **then**
7         Group and Refill the selected
          columns as well as rows to a
          dense block, and update $m$
8       **end**
9     **end**
10   **end**
11 **end**
12 Set other elements out of dense blocks to 0

---

[a] http://glaros.dtc.umn.edu/gkhome/metis/hmetis/overview

Here we provide a detailed description of how many and which channels we choose to refill. Our main experiments adopt the $\ell_1$ norm of channel weights as the picking criterion to score the channel importance due to its superior performance. Let $\theta^l \in \mathbb{R}^{c_{\text{out}} \times n}$ denotes the parameters of the convolutional layer $l$, where $c_{\text{out}}$ is the number of output channel and $n$ is the continued product of the number of input channel, channel height and weight, as shown in Figure 2. $\theta_i^l \in \mathbb{R}^n$ represents the weights in the $i$th kernel and $m_i^l \in \{0, 1\}^{|\theta_i^l|}$ is the corresponding mask. We first calculate the $\ell_1$ norm of $m_i^l \odot \theta_i^l$, which is a summation of the absolute value of remaining weights in the kernel $i$. Then we use it to pick the top-$k$ scored kernels, which will be fully refilled. $k = \lceil s^l \times c_{\text{out}} \times n \rceil$, where $s^l$ is the original layerwise sparsity and $c_{\text{out}} \times n$ is the total number of weights in kernel $i$. Meanwhile, the rest $c_{\text{out}} - k$ kernels are dropped for efficiency gains.

Furthermore, we propose a soft version, *refilling+*, to make a redemption for the aggressive nature of wiping out all remaining channels. It picks and re-actives an extra proportion of channels to slow down the network capacity reduction, as indicated by shallow blue blocks in Figure 2.

### 3.3 REGROUPING FOR STRUCTURAL PATTERNS

Although proposed *refilling(+)* reorganizes the unstructured mask and produces useful channel-wise structural subnetworks, it is rigid and inelastic since the smallest manageable unit is a kernel. In other words, the dense matrices in identified structural patterns have a restricted shape where one dimension must align with the kernel size $n$, i.e., the continued product of the number of input channels, channel height and weight. Motivated by Rumi et al. (2020), we introduce a *regrouping* strategy (Figure 2) to create more fine-grained group-wise structural patterns with flexible shapes for remaining dense matrices.

▷ **How to perform regrouping?** *Regrouping* aims to find and extract dense blocks of non-pruned elements in the sparse weight matrix. These blocks have diverse shapes, as demonstrated in Figure 2, which are usually smaller in size compared to the original sparse matrix. Note that a channel/kernel can be regarded as a special case of the dense block.

As described in Algorithm 3, to achieve the goal, we first need to find similar rows and columns, and then bring them together. Specifically, We adopt the Jaccard similarity (Rumi et al., 2020; Jiang et al., 2020) among non-zero columns as the similarity between two rows in the sparse matrix, which is calculated as a cardinality ratio of the intersections to the union of non-zero columns. For instance, kernel 1 and kernel 2 in Figure 2 (upper left) share three columns in eight non-zero distinct columns, and their similarity is $\frac{3}{8}$. Then, if two rows have a larger similarity, it can form a denser block when we group them together. Take Figure 2 as an example. We can group kernel $1, 2, 3$'s non-zero columns $1, 3, 6, 11$ with at least two elements together, which leads to the first orange dense block.

More precisely, we take the hypergraph partitioning in the regrouping algorithm to generate dense blocks. It treats each row and column from the sparse matrix as a node and hyperedge in the hypergraph, where hyperedge (i.e., column) connects the corresponding nodes (i.e., row). Then, the pairwise similarity is leveraged to locate an optimal partitioning, which can be achieved with hMETIS[2]. More details are referred to Rumi et al. (2020). After obtaining the desired dense blocks, we enable all their parameters to be trainable by refilling the corresponding pruned elements. Note that refilling these pruned weights does not cause any efficiency loss since the size of the blocks is fixed, while it potentially maximizes the usage of these blocks and brings accuracy gains. Meanwhile, the rest parameters not included in the dense blocks will be discarded, i.e., setting the corresponding position in binary mask $m$ to zero, for reducing the computational overhead as illustrated in Figure 2. It is because any parameters outside the dense blocks require extra weights loading and have little data reuse (Rumi et al., 2020), which harms the trade-off of accuracy and efficiency.

▷ **How refilled / regrouped dense blocks be beneficial?** We notice that the common tools like cuDNN (Chetlur et al., 2014) have a significant drawback that the inference time does not linearly change with the number of kernels, since they are only optimized for kernel matrices with a multiple of 32 rows (Radu et al., 2019). For example, as stated in Rumi et al. (2020), a convolutional layer with 10 kernels might have a similar inference time with a convolutional layer with 32 kernels. However, the number of kernels in these dense blocks is almost arbitrary, so a more sophisticated GEMM-based efficient implementation (Rumi et al., 2020) is needed to accelerate better our refilled / regrouped structural patterns. Following Rumi et al. (2020), we split a kernel with $r$ rows into two parts: one has $\lceil r/32 \rceil \times 32$ rows and the other one has $r \bmod 32$ rows. First, we directly apply the standard GEMM-based convolution algorithm with shared memory to cache the input and output matrix. For the second part, due to the poor data reuse of input matrices, we choose caching the kernel and output matrices for an improved cache hit rate and overall performance. More details are referred to Rumi et al. (2020).

## 4 THE EXISTENCE OF STRUCTURAL WINNING TICKETS

**Tiny-ImageNet and ImageNet.** In this section, we reveal the existence of our proposed structural winning tickets on ImageNet and Tiny-ImageNet with ResNet-50 backbone. Results of unstructured `IMP`, channel-wise structural `IMP-Refill(+)`, and group-wise structural `IMP-Regroup` are collected in the Figure 3. The end-to-end inference time[3] of obtained structural winning tickets with

---

[2]http://glaros.dtc.umn.edu/gkhome/metis/hmetis/overview
[3]TorchPerf (https://github.com/awwong1/torchprof) is adopted as our tool to benchmark both the end-to-end and layer-wise running time on GPU devices.

extreme sparsity levels are presented, which is calculated on a single 2080 TI GPU with a batch size of $64$. Extreme sparsity is defined as maximum sparsity when the subnetwork has superior accuracy than its dense counterpart.

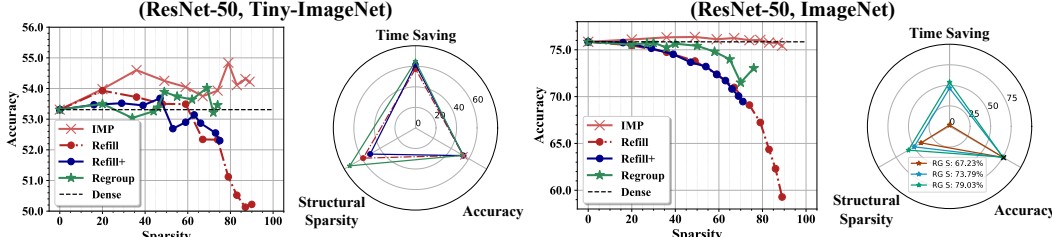

Figure 3: (*Curve plots*) Testing accuracy (%) over network sparsity levels (%) on Tiny-ImageNet and ImageNet datasets with ResNet-50 (25.56 M). (*Radar plots*) The end-to-end inference time saving of extreme structural winning tickets. Note that unstructured subnetworks or dense models do not have structural sparsity, and thus they are plotted as dots in the axes of accuracy in the corresponding radar plot. The rightmost plot includes three extreme regroup tickets with accuracy drop $< 1\%$, where "RG S: $x\%$" indicates unstructured sparsity before regrouping.

From Tiny-ImageNet results in Figure 3 (*left*), several positive observations can be drawn: ❶ Structural winning tickets with $60\%$ channel-wise structural sparsity and $74\%$ group-wise structural sparsity are located by `IMP-Refill` and `IMP-Regroup` respectively, which validate the effectiveness of our proposals. ❷ Although at the high sparsity levels (i.e., $> 50\%$), `IMP-Refill+` outperforms `IMP-Refill` if they are from the same unstructured IMP subnetworks. Considering the overall trade-off between channel-wise structural sparsity and accuracy, `IMP-Refill` appears a clear advantage. A possible explanation is that *refilling+* seems to bring undesired channels which potentially result in a degraded performance trade-off. ❸ `IMP-Regroup` performs better at high sparsities. It is within expectation since fine-grained group-wise structural patterns tend to make the networks be more amenable to pruning. ❹ Extreme channel- / group-wise structural winning tickets with $45\% \sim 50\%$ / $74\%$ sparsity from `IMP-Refill(+)` / `IMP-Regroup` achieve $57.53\% \sim 61.79\%$ / $64.84\%$ GPU running time savings, without sacrificing accuracies.

As for large-scale ImageNet experiments, the conclusion are slightly different: ❶ There is almost no difference between the performance of `IMP-Refill` and `IMP-Refill+`, and both can not find channel-wise structural winning tickets. But it seems to suggest our picking rule (i.e., channel weights' $\ell_1$ norm) provides a great estimation for channel importance, although it is too aggressive for ImageNet experiments. ❷ The group-wise structural winning ticket at $31\%$ sparsity is still exist in (RN-50, ImageNet), while the low sparsity brings limited $1\%$ time savings. For a better efficiency and performance trade-off, `IMP-Regroup` is capable of locating structural subnetworks at $51\%$ / $58\%$ sparsity with $53.75\%$ / $64.84\%$ time savings and $0.33\%$ / $0.95\%$ accuracy drop.

**CIFAR with diverse network architectures.** We then validate our approaches on CIFAR-10/100 (C10/100) with diverse network backbones including Wide-ResNet-32-2, MobileNet-v1, VGG-16, and ResNet-18. Based on the extensive results in Figure 4 and 5, we find: ❶ On {(WRN-32-2,C10), (WRN-32-2,C100), (MBNet-v1,C10), (MBNet-v1,C100), (VGG-16,C10), (VGG-16,C100), (RN-18,C10), (RN-18,C100)} schemes, we consistently disclose the existence of structural winning tickets with {$53\%$, $28\%$, $67\%$, $0\%$, $60\%$, $40\%$, $50\%$, $0\%$} channel-wise sparsity and {$66\%$, $36\%$, $72\%$, $56\%$, $80\%$, $80\%$, $78\%$, $78\%$} group-wise sparsity from `IMP-Refill(+)` and `IMP-Regroup`, respectively. ❷ With the same network, pursuing channel-wise sparse patterns on CIFAR-100 is more challenging than it on CIFAR-10, possibly due to the larger dataset complexity. On the same dataset, larger networks tend to have larger extreme sparsities for both channel- and group-wise structural winning tickets, with the exception of `IMP-Refill(+)` on (RN-18, C100). ❸ At the middle sparsity levels (i.e., $< 50\%$), `IMP-Regroup` behaves closely to `IMP-Refill(+)`, while `IMP-Regroup` has a superior performance at high sparsity levels. ❹ Up to {$57.75\%$, $60.60\%$, $55.45\%$, $64.93\%$} GPU running time savings are obtained by group-wise structural winning tickets with undamaged performance on {(VGG-16,C10), (VGG-16,C100), (RN-18,C10), (RN-18,C100)}, which surpass `IMP`, `IMP-Refill(+)`, and dense models by a significant efficiency margin. A exception is that `IMP-Refill` on (VGG-16,C10) achieves the best time savings, i.e., $63.11\%$.

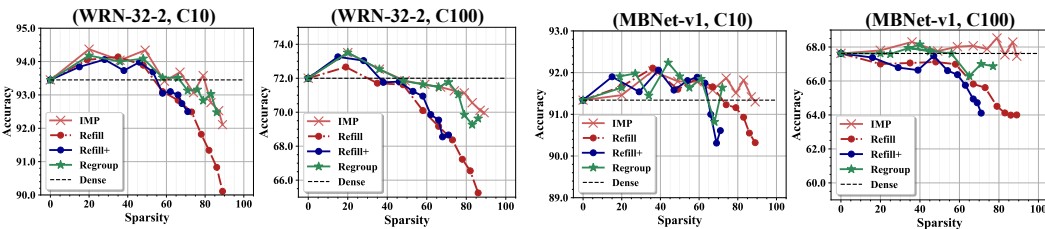

Figure 4: Testing accuracy (%) over network sparsity levels (%) on CIFAR-10/100 with small models Wide-ResNet-32-2 (1.86 M) and MobileNet-v1 (3.21 M).

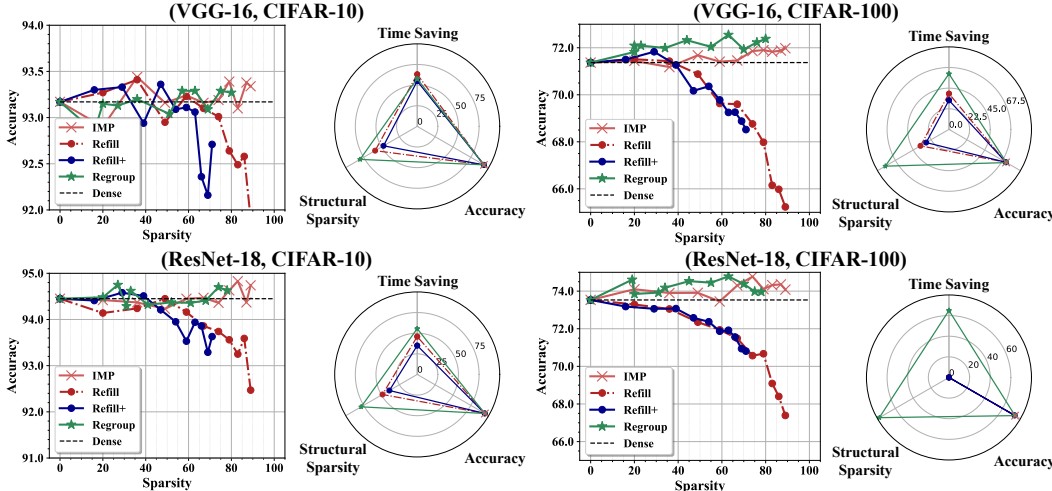

Figure 5: (*Curve plots*) Testing accuracy (%) over network sparsity levels (%) on CIFAR-10/100 with large models VGG-16 (14.72 M) and RN-18 (11.22 M). (*Radar plots*) The end-to-end inference time saving of extreme structural winning tickets. Note that unstructured subnetworks or dense models do not have structural sparsity, and thus they are plotted as dots in the axes of accuracy in the corresponding radar plot.

**Layer-wise speedups.** Figure 6 shows the layer-wise speedup performance of convolution operations in VGG-16's extreme structured winning tickets from different algorithms. `IMP-Regroup` presents impressive layer-wise speedups up to 6.67x compared to others, especially on the last a few layers (e.g., conv. 12). The possible reasons lie in two aspects: (*i*) the latter layers reach a larger compression ratio and have greater potentials for acceleration; (*ii*) the *regrouping* algorithm prefers convolutional layers (i.e., latter layers in VGG-16) with a larger number of kernels which benefits to group appropriate dense blocks, as also suggested by Rumi et al. (2020).

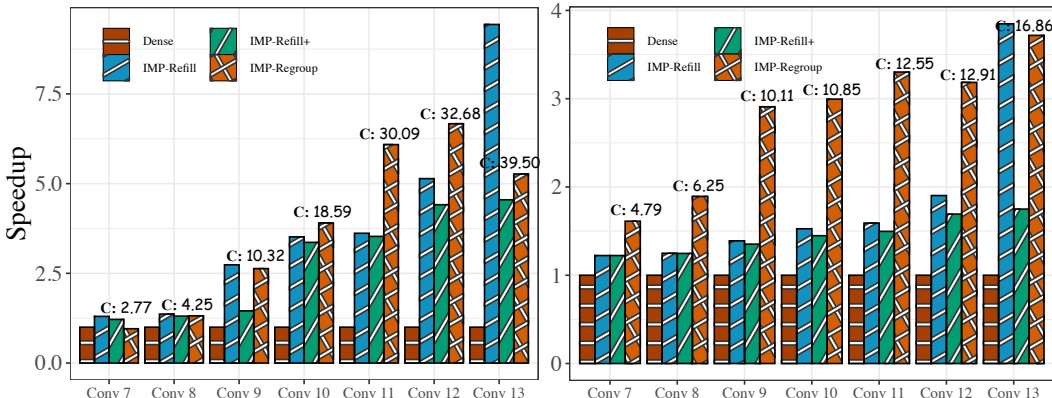

Figure 6: The layer-wise performance of convolution operations in extreme structural winning tickets of (VGG-16, C10). The first six conv. operations are omitted since there is no meaningful speedup, coincided with Rumi et al. (2020). Marks like "C: 2.77" indicate the layer-wise compression ratio of `IMP-Regroup`.

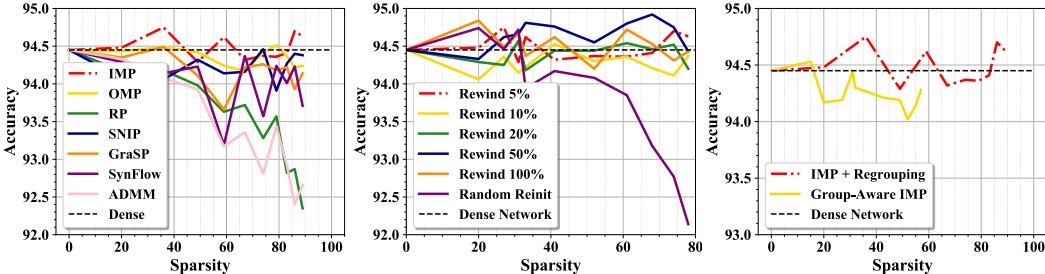

Figure 7: (*Left*) Performance of structural tickets grouped from diverse initial unstructured masks. (*Middle*) Performance of group-wise structural tickets with different weight rewinding. (*Right*) Performance comparisons between `IMP-Regroup` and group-aware `IMP` as described in Algorithm 4. Testing accuracies (%) over network sparsity levels (%) are reported on (RN-18,C10).

## 5 ABLATION STUDY AND VISUALIZATION

**Different sources of unstructured masks.** Intuitively, the initial unstructured sparse mask should plays an essential role in the achievable performance of our proposed "post-processing techniques". We therefore conduct a comprehensive ablation study about the various sources of the initial sparse masks in Figure 7, including `IMP`, `OMP`, `RP`, `SNIP`, `GraSP`, `SynFlow`, and `ADMM`. The details of comparison methods are included in Section 3.1. We observe that `IMP` and `OMP` provide initial unstructured masks with the top-2 highest quality for our *regrouping* algorithm, in terms of the train-from-scratch accuracy of grouped structural subnetworks.

**Different initialization for the re-training.** Initialization (Frankle & Carbin, 2019; Renda et al., 2020) as another key factor in LTH, also contributes significantly to the existence of winning tickets. To exhaustively investigate the effect from different initialization (e.g., rewound weights), we launch experiments started from diverse rewound weights ({5%, 10%, 20%, 50%, 100%} of total training epochs) as well as a random re-initialization. As shown in Figure 7, using 50% rewound weight reach the overall best performance; other weight rewinding setups perform similarly and clearly surpass random re-initializing at sparsity levels > 30%.

**Group-aware `IMP`** This work mainly focuses on the post-processing of unstructured sparse masks. Another possibility is integrating *regrouping* into `IMP` by alternatively performing unstructured magnitude pruning and regrouping, which we term as group-aware `IMP`. From Fig. 7, it has a worse performance due to the stricter constraint on sparse patterns, compared to `IMP-Regroup`.

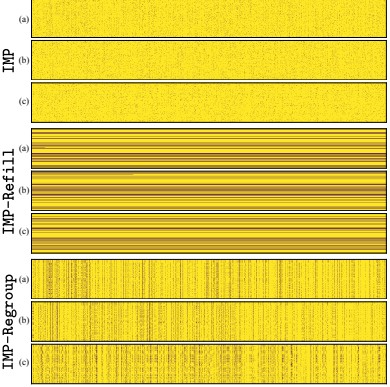

Figure 8: Sparse mask visualizations of the extreme winning tickets from `IMP` (unstructured), `IMP-Refill` (channel-wise structural), and `IMP-Regroup` (group-wise structural) on (VGG-16,C10). The darker color indicates the remaining unpruned elements. (a,b,c) are the last three conv. layers.

**Visualization of sparse masks.** Figure 8 visualizes different types of obtained sparse masks from (VGG-16,C10). Sub-figures (a,b,c) plot the mask matrices of size $c_{out} \times n$ for certain layers. Similar to the illustration in Figure 2, `IMP-Refill` masks show clear kernel-wise sparse patterns across the rows, and `IMP-Regroup` masks present fine-grained structural sparse patterns capable of forming neat dense blocks after regrouping.

## 6 CONCLUSION

In this paper, we challenge the "common sense" that an identified `IMP` winning ticket can only have unstructured sparsity, which severely limits its practical usage due to the irregular patterns. We for the first time demonstrate the existence of structural winning tickets by leveraging post-processing techniques, i.e., *refilling(+)* and *regrouping*. The located channel- and group-wise structural subnetworks achieve significant inference speedups up to 6.67x on hardware platforms. In this sense, our positive results bridge the gap between the lottery ticket hypothesis and practical accelerations in real-world scenarios. We would be interested in examining LTH with more effective structural sparsity for real-time mobile computing in future work.

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

## A1   MORE IMPLEMENTATION DETAILS

**Group-aware `IMP`**   Here we provides the detailed procedures of group-aware `IMP` in Algorithm 4. Intuitively, it embeds *regrouping* (Algorithm 3) into `IMP` (Algorithm 1) by performing *regrouping* on the unstructured mask $m$ from each `IMP` round.

---
**Algorithm 4:** Group-aware `IMP`

---
**Input:** $f(x; \theta_0)$, group-wise structural sparsity $s$
**Output:** $f(x; m \odot \theta_i)$ with group-wise structural sparse mask $s$

1  Set the pruning mask $m = \mathbf{1} \in \mathbb{R}^{|\theta|}$
2  Train $f(x; \theta_0)$ to rewinding step $i$: $f(x; \theta_i) = \mathcal{A}_i^{\mathcal{T}}(f(x; \theta_0))$
3  **while** *not reach sparsity $s$* **do**
4  $\quad$ Train $f(x; m \odot \theta_i)$ to step $t$: $f(x; m \odot \theta_t) = \mathcal{A}_{t-i}^{\mathcal{T}}(f(x; m \odot \theta_i))$
5  $\quad$ Pruning 20% of remaining weight of $m \odot \theta_t$, and update $m$
6  $\quad$ Refining the unstructured mask $m$ by performing *regrouping*, as shown in Algorithm 3
7  **end**

---

**Profiling.**   To compute the GPU running time of regrouped convolution layers, we adopt their CUDA C/C++ implementation. Our results do not include the running time of normalization and activation layers, following the standard in Rumi et al. (2020). For a fair calculation, we feed the *same* input features to convolution layers that belong to the same model. For ResNet-18 and VGG-16, the size of the input features is $(64, 64, 127, 127)$. For ResNet-50, the size of input features is $(64, 64, 64, 64)$. The GPU we use for profiling is NVIDIA RTX 2080 TI, with a CUDA version of 10.2 and a cuDNN (Chetlur et al., 2014) version of 7.6.5.

## A2   MORE EXPERIMENT RESULTS

**Different channel picking criterion for refilling.**   We ablation the channel picking criterion for `IMP-Refill(+)`, including ❶ the $\ell_1$ norm of channel's remaining weight, ❷ the $\ell_1$ or $\ell_2$ norms of channel's feature map, ❸ the number of remaining weights in the channel, ❹ the channel's saliency score (Molchanov et al., 2019). Experiment results are collected in Figure A9, which demonstrate the superior performance of `IMP-Refill` w. $\ell_1$ of channel weights (yellow curve).

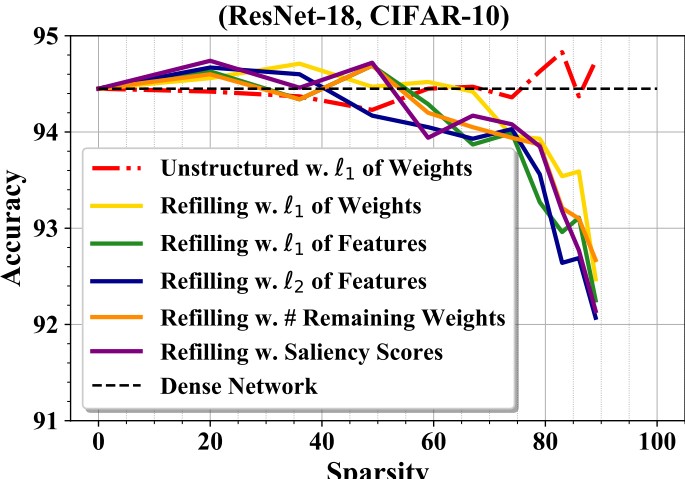

Figure A9: Performance of structural tickets refilled by diverse channel picking criterion. Testing accuracies (%) over network sparsity levels (%) are reported on (RN-18,C10).

**All sections of newly added results and discussions are highlighted.**

## A3   APPLICATION TO OTHER TICKETS

We further study whether our methods rely on types of random initialization. We first identify tickets in ResNet-50 on ImageNet, and then transfer the tickets we have found to CIFAR-10. The test accuracy of the un-pruned (dense) ResNet-50 on CIFAR-10 is 95.37%. Multiple results from `IMP-Refill IMP-Regroup` and are reported in Table A2. We can see from Table A2 that `IMP-Refill` can locate structured winning tickets at the sparsity around 36%, and `IMP-Regroup` can locate structured winning tickets at higher sparsity (more than 56.00%). These results suggest that our refill and regroup method can work on another kind of tickets (*i.e.,* pre-training tickets)

Table A2: Testing accuracy and percentage of remaining weights on ResNet-50 with CIFAR-10. Different methods (`IMP-Refill`, and `IMP-Regroup`) are evaluated. The baseline (test accuracy of the dense network) is 95.37%.

| IMP-Refill | | IMP-Regroup | |
|---|---|---|---|
| Remaining Weight | Accuracy | Remaining Weight | Accuracy |
| 64.14% | 95.81 | 77.28% | 94.94 |
| 51.37% | 95.14 | 71.40% | 96.28 |
| 41.01% | 94.51 | 67.60% | 94.09 |
| 32.76% | 94.38 | 59.43% | 95.65 |
| 26.17% | 94.19 | 51.84% | 95.39 |
| 20.97% | 94.11 | 43.99% | 95.51 |

## A4   RESULTS UNDER DIFFERENT TRAINING SETTINGS

To provide more experimental results, we launch experiments under different settings with VGG-16, WideResNet-32-2, and ResNet-50. The changes of training settings are summarized below:

1. For VGG-16, we increase the number of training epochs to 240, and we decay the learning rate at 150th, 180th, and 210th epoch.

2. For WideResNet-32-2, we did not split the official training set into the a training and a validation set as other experiments did. We also report the best validation accuracy instead of the best test accuracy. We also increase the number of training epochs to 240 and decay the learning rate at 150th, 180th, and 210th epoch.

3. For ResNet-50, we replace the first convolution layer to be of kernel size 3, padding size 1, and strides 1.

**VGG-16 on CIFAR-100.**   The proportion of remaining weights and testing accuracy of (VGG-16,C100) are shown in Table A3. From the table we can see that our conclusions still hold: `IMP-Regroup` can locate structured winning tickets at very high sparsity ($> 75\%$).

**WideResNet-32-2 on CIFAR-100.**   The proportion of remaining weights and testing accuracy of (WRN-32-2,C100) are shown in Table A4. From the table we can see that our conclusions still hold: `IMP-Regroup` can locate structured winning tickets at about 75% sparsity, and `IMP-Refill` can locate structured winning tickets at 20% sparsity.

**ResNet-50 on Tiny-ImageNet.**   The proportion of remaining weights and testing accuracy of (RN-50, Tiny-ImageNet) are shown in Table A5. From the table we can see that our conclusions still hold: `IMP-Regroup` can locate structured winning tickets at about 42% sparsity, and `IMP-Refill` can locate structured winning tickets at 20% sparsity.

Table A3: Testing accuracy and percentage of remaining weights on CIFAR-100 with VGG-16. Different methods (`IMP`, `IMP-Refill`, and `IMP-Regroup`) are evaluated. The baseline (test accuracy of the dense network) is 73.43%.

| Round | IMP | | IMP-Refill | | IMP-Regroup | |
|---|---|---|---|---|---|---|
| | Remaining Weight | Accuracy | Remaining Weight | Accuracy | Remaining Weight | Accuracy |
| 1 | 80.00% | 73.64 | 80.17% | 73.43 | 82.36% | 73.63 |
| 2 | 64.00% | 73.80 | 64.06% | 72.87 | 80.00% | 73.81 |
| 3 | 51.20% | 73.67 | 51.31% | 72.67 | 69.46% | 74.31 |
| 4 | 40.96% | 74.01 | 41.08% | 71.37 | 62.61% | 73.94 |
| 5 | 32.77% | 74.27 | 32.85% | 70.79 | 56.09% | 75.05 |
| 6 | 26.21% | 74.56 | 26.33% | 71.07 | 46.53% | 74.98 |
| 7 | 20.97% | 74.58 | 21.03% | 69.42 | 38.18% | 75.24 |
| 8 | 16.78% | 74.52 | 16.94% | 68.75 | 30.98% | 74.68 |
| 9 | 13.42% | 74.42 | 13.42% | 67.25 | 25.27% | 75.25 |

Table A4: Testing accuracy and percentage of remaining weights on CIFAR-100 with WideResNet-32-2. Different methods (`IMP`, `IMP-Refill`, and `IMP-Regroup`) are evaluated. The baseline (validation accuracy of the dense network) is 75.53%.

| Round | IMP | | IMP-Refill | | IMP-Regroup | |
|---|---|---|---|---|---|---|
| | Remaining Weight | Accuracy | Remaining Weight | Accuracy | Remaining Weight | Accuracy |
| 1 | 80.00% | 76.21 | 80.00% | 75.46 | 80.00% | 75.98 |
| 2 | 64.00% | 75.78 | 64.06% | 74.59 | 64.00% | 76.19 |
| 3 | 51.20% | 76.02 | 51.51% | 73.53 | 51.20% | 76.13 |
| 4 | 40.96% | 75.92 | 41.51% | 72.95 | 40.96% | 75.88 |
| 5 | 32.77% | 76.07 | 32.84% | 72.12 | 32.99% | 75.98 |
| 6 | 26.21% | 75.74 | 26.53% | 70.91 | 26.27% | 75.78 |
| 7 | 20.97% | 75.92 | 21.11% | 69.55 | 21.76% | 74.74 |
| 8 | 16.78% | 75.87 | 17.11% | 67.74 | 18.14% | 73.85 |
| 9 | 13.42% | 75.41 | 13.67% | 65.73 | 14.85% | 72.99 |

Table A5: Testing accuracy and percentage of remaining weights on Tiny-ImageNet with ResNet-50 (second implementation). Different methods (`IMP`, `IMP-Refill`, and `IMP-Regroup`) are evaluated. The baseline (test accuracy of the dense network) is 65.33%.

| Round | IMP | | IMP-Refill | | IMP-Regroup | |
|---|---|---|---|---|---|---|
| | Remaining Weight | Accuracy | Remaining Weight | Accuracy | Remaining Weight | Accuracy |
| 1 | 80.00% | 65.44 | 80.30% | 65.27 | 80.15% | 65.51 |
| 2 | 64.00% | 65.69 | 64.16% | 63.40 | 68.25% | 65.16 |
| 3 | 51.20% | 65.50 | 51.42% | 61.89 | 58.19% | 65.21 |
| 4 | 40.96% | 65.73 | 41.08% | 60.43 | 54.19% | 64.42 |
| 5 | 32.77% | 65.23 | 32.85% | 59.64 | 51.75% | 64.52 |

## A5 FLOPs Results

We calculate the FLOPs of VGG-16 on CIFAR-10 processed with different methods. The FLOPs of the dense VGG-16 is about 0.314G. We select models with similar sparsity for better comparison. The sparsities of IMP-Refill, IMP-Refill+, and IMP-Regroup are $\{32.84\%, 46.41\%, 20.12\%\}$, respectively, and the FLOPs are $\{0.089G, 0.122G, 0.093G\}$, respectively. It is noteworthy that Refill and Refill+ will lower the input and the output channel of a convolution layer while Regroup cannot, so Refill and Refill+ can save more FLOPs under the same level of sparsity.

## A6 Other Visualization

From Figure A10 we can see that, similar to `IMP-Refill`, `IMP-Refill+` masks also show kernel-wise sparse patterns across the rows.

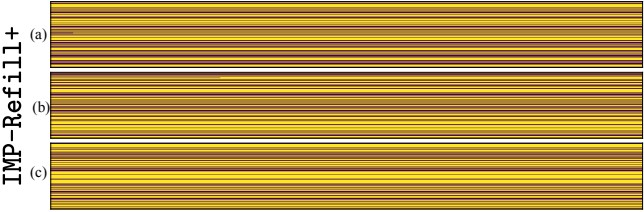

Figure A10: Sparse mask visualizations of the extreme winning tickets from `IMP-Refill+` (channel-wise structural), and on (VGG-16,C10). The darker color indicates the remaining unpruned elements. (a,b,c) are the last three conv. layers.

Except radar plots, we also include histograms to demonstrate the triads - Sparsity, Speedup, and Accuracy, which are shown in Figure A11. In each histogram, we report the metrics of four methods: `Dense`, `IMP-Refill`, `IMP-Refill+`, and `IMP-Regroup`. `Dense` has no sparsity and no speedup, so the corresponding bars are always unseen from the charts.

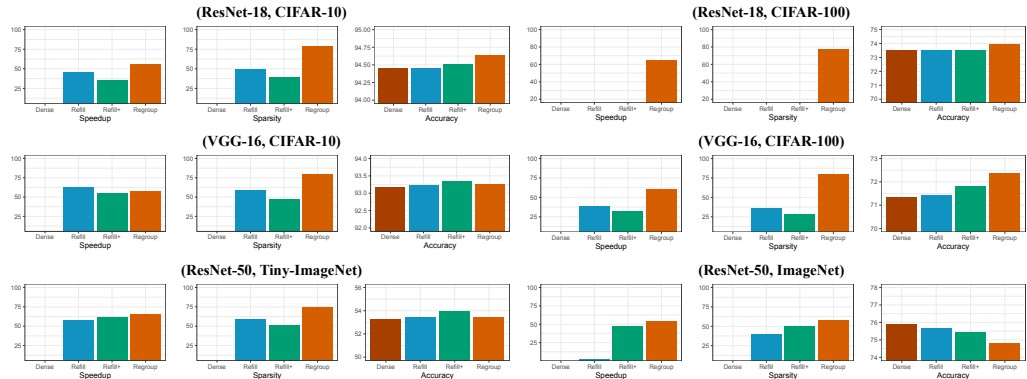

Figure A11: The sparsity, speedup, and test accuracy of various models (ResNet-18, VGG-19, ResNet-50) on various datasets (CIFAR-10/-100, Tiny ImageNet, ImageNet).

