# OpenReview forum: "Lottery Tickets can have Structural Sparsity"
_ICLR.cc/2022/Conference — ICLR 2022 Submitted_

### Official Review · Reviewer_iox5 · 2021-10-31

**Correctness:** 4
**Technical Novelty And Significance:** 3
**Empirical Novelty And Significance:** 4
**Recommendation:** 8
**Confidence:** 4

**Main Review:**

It is well known that lottery tickets so far are only successfully found with unstructured pruning. This paper challenges this commonsense, and its positive finding could be of broad interest and impact. Instead of directly adopting structural pruning, the authors propose the postprocessing techniques to accompany IMP, namely, the refilling/regrouping algorithm to form channel-wise and group-wise structural sparsity, respectively. These algorithms are simple and intuitive, yet novel and convincing.  The paper for the first time achieves high sparsity (up to 80%) with structure in the lottery ticket literature and shows practical GPU hardware runtime savings at around 50-60%.

I suggest the authors to tone down their claims. There definitely exist similar works in relevant problem domains, such as sparse training with structures: for example, “Learning N:M Fine-grained Structured Sparse Neural Networks From Scratch” ICLR 2021, should be discussed and compared.
Besides the scientific interest, I am not sure if I understand from a practical point of view, why we really need particularly pursue a structurally sparse lottery ticket, in contrast to directly using a structurally pruned NN.
Also, how this work compares to using NAS to search for channel numbers?
I don't understand for ImageNet, why the current results can show "our picking rule (i.e., channel weights l1 norm) provides a great estimation for channel importance"
No theoretical insight on why the proposed algorithm can work is provided. But for an empirical pilot study, this is fair and acceptable.


**Summary Of The Paper:**

This work demonstrates the existence of structural winning tickets for the first time by leveraging post-processing techniques.

**Summary Of The Review:**

The contribution of this paper is solid. I hope the authors could clarify the weak points.

---

> ### Author Response · Authors · 2021-11-19
> **Response to Reviewer iox5**
>
> Thank you for appreciating the novelty and the simplicity of our method! Your comments are important to us and we have addressed them all below.
>
> **[Cons 1: Related work]**
>
> Thank you for pointing out this important work in the literature. We will include the discussion on this work and compare it with our methods. They focus on exploring structural pruning methods combining the advantage of unstructured sparsity. Their investigations are parallel to our LTH settings and problems since they propose to learn a fine-grained sparse network at training time, rather than extracting the sparse subnetwork as LTH does.
>
> It is an interesting work and we would be definitely interested to try it as future works. Meanwhile, we will more specify our claims to be within the realm of LTH according to your suggestion.
>
> **[Cons 2: The meaning of structured LTH]**
>
> We are glad that you recognize the scientific merit of finding structured lottery tickets. Its practical relevance, in comparison to general structural prunings, could manifest in at least two more points:
>
> (1) as presented in our response to reviewer TFc4, our method can also apply to finding the transferable lottery ticket from a pre-trained model. That is well-known as a promising practical application of LTH to generate “universal subnetworks” that can be memory and computation-efficient to transfer. Our structured mask further brings hardware resource benefits into this regime, for the first time.
>
> (2) our proposed techniques have a plug-and-play “postprocessing” nature, and can also be leveraged to convert other general unstructured pruning methods into structured versions as well, beyond LTH. That could enable a more flexible and delicate trade-off between accuracy and efficiency, which is beyond the scope of this LTH-focused paper but will be explored in future work.
>
> **[Cons 3: The connection with NAS]**
>
> NAS is indeed similar to channel pruning in finding optimal channel configuration in models. However, in our work, we find that the channel-wise sparsity does not bring much performance benefit compared to the fine-grained structural sparsity. To the best of our knowledge, NAS currently cannot find such fine-grained structural sparsity. Besides, the main value in LTH, i.e., to find extremely sparse, yet trainable and performant neural networks, is relevant yet not fully overlapped with NAS.
>
> **[Cons 4: Why our method can estimate channel importance]**
>
> Thank you for your suggestion. We will explain our reasoning process below, and we will rephrase this sentence for better understanding at the same time.
>
> We use the l1 norm of weights to rank channels, then we keep channels with the largest norms. Refill+ picks more channels than Refill, but on ImageNet the two methods have similar performance. In another word, the extra channels picked by Refill+ contribute less or nearly no to the test accuracy. This also acts as evidence that Refill provides a great estimation of channel importance - the channels picked by Refill seem to be both essential and sufficient for retaining performance. We have also clarified this point in the revised manuscript.

---

> > ### Comment · Reviewer_iox5 · 2021-11-25
> > **Post rebuttal responses**
> >
> > I am satisfied with the authors‘ rebuttal and agree with the other positive reviewers. I recommend acceptance of this paper.

---

> > > ### Author Response · Authors · 2021-11-26
> > > **Response to Reviewer iox5**
> > >
> > > Dear Reviewer **iox5**,
> > >
> > > We truly thank reviewer **iox5** for highly acknowledging our paper as novel, solid, and convincing. We really appreciate reviewer iox5 for increasing our score and supporting the acceptance of our paper.
> > >
> > > We are again very thankful for your time and all constructive feedback!
> > >
> > > Best wishes,
> > >
> > > Authors

---

### Official Review · Reviewer_YWUh · 2021-11-02

**Correctness:** 1
**Technical Novelty And Significance:** 2
**Empirical Novelty And Significance:** 2
**Recommendation:** 5
**Confidence:** 5

**Main Review:**

Strengths

●	Up to date, LTH has not been shown valid on structured pruning (I mean, filter pruning or channel pruning). This paper is meant to bridge this gap, which could be a significant step forward.

●	They propose several post-processing techniques, which deliver practical speedup. Potentially, they can be very useful.

Weaknesses

1.	One contribution this paper keeps emphasizing is that they find structural winning tickets for the first time. However, some important concepts, baseline results are either not established or (potentially) flawed. I do not think the presented results can faithfully support their claimed contribution.

First, one important property of LTH is that, if the subnet is trained using randomly re-initialized weights instead of the winning tickets, the subnet cannot achieve full accuracy. This paper does not have this kind of comparison at all. For example, Fig. 3, for channel sparsity (refill and refill+, not regroup), what are the accuracies when you train the pruned network from scratch (I mean, fully randomly reinit all the weights in the pruned network)? Without this comparison, we cannot tell if the initial weights can be identified as “winning tickets” in the first place.

Second, the presented results are (potentially) flawed. Many baseline accuracies are far below the average reported by other papers. E.g., in Fig. 3, ResNet50 on TinyImageNet, others typically can get ~67% accuracy (see [1*]), while this paper only got a little bit over 53% at sparsity 0. More than 10% gap.

Similar issue: Fig 3, ResNet50 on ImageNet. The typical top-1 accuracy is around 76.1% (see torchvision models), while this paper reports 75.2%. Nearly 1 point top-1 accuracy drop on ImageNet is quite big. Fig 4, their WRN-32-2 on CIFAR100 only achieves 72%, while even WRN-16-2 (much smaller than WRN-32-2) can achieve 73.26% (see [2*]). Fig 5, their VGG16 on CIFAR100 only archives less than 72%, while VGG13 can achieve over 74% (see [2*]).

I thus deeply doubt the validity of these experiments -- Note, the accuracy of dense models is important in validating LTH because it is the baseline accuracy that subnets are going to compare against. Using a lower baseline accuracy, of course it would be easy to achieve high sparsity “without performance drop”.

-- I think this is also why in many plots, the accuracy of pruned models can strangely surpass their dense counterparts even at a large sparsity.

2.	They do not really achieve what they claimed.

In pruning literature, structured pruning typically narrowly means filter or channel pruning,  while this paper actually also refers to a more fine-grained sparsity structure (see the “Group-wise Structural Sparse Mask” in Fig. 2). This actually make it much less interesting, because this kind of sparsity still needs customized implementation (“a more sophisticated GEMM-based efficient implementation (Rumi et al., 2020) is needed to accelerate better our refilled / regrouped structural patterns”). How to get practical speedup is actually unknown from this paper (they defer many important details to (Rumi et al., 2020)).

Then, only look at the channel sparsity. Fig. 3, ResNe50 on ImageNet, Refill and Refill+ start to drop accuracy even at a marginal sparsity. This means they do not find winning tickets. They also admitted this in the paper: “There is almost no difference between the performance of IMP-Refill and IMP-Refill+, and both can not find channel-wise structural winning tickets”.

What attracted me at the first sight of this paper is that they validate LTH on filter/channel pruning. Unfortunately, they do not really achieve what they claimed.

3.	From the methodology perspective, the proposed techniques are “post-processing”. Most structured pruning methods are independent from unstructured pruning. While here, they blend them together with “post-processing” instead of proposing a really independent technique to find winning tickets for filter/channel sparsity. I would say, it doesn't look like an elegant neat scheme (although “elegant” may not be really important).

4.	Minor issue. Presentation: Fig. 3, radar plots. It is pretty hard to read the radar plots. In Particular, some dots are overlapping and pretty small. They are barely recognizable in print. They can be presented in a better way.

[1*] https://openaccess.thecvf.com/content_CVPR_2020/papers/Yuan_Revisiting_Knowledge_Distillation_via_Label_Smoothing_Regularization_CVPR_2020_paper.pdf

[2*] https://arxiv.org/pdf/1910.10699.pdf, ICLR 2020


**Summary Of The Paper:**

LTH (lottery ticket hypothesis) was mainly proposed for unstructured pruning. This paper shows it can also be validated on structured pruning, for the first time. The key for them to achieve so is the newly proposed post-processing techniques, refilling(+) and regrouping. They show by these techniques, structural winning tickets can be found, with up to 6.67x on hardware platforms.

**Summary Of The Review:**

Important baseline results are missing. Experiments are potentially flawed. They claimed what is not really realized in the paper.

---

> ### Author Response · Authors · 2021-11-19
> **Response to Reviewer YWUh (Part 1)**
>
> Thanks for your valuable comments on our paper! We provide you with detailed point-to-point answers below. We hope our response can address your concerns.
>
> **[Cons 1: Training from scratch with re-initialization weights]**
> Thank you for your suggestion. We have included the performance of training from scratch with randomly re-initialization weights below. ResNet-18 on CIFAR-10 is adopted for both IMP-Refill and IMP-Regroup experiments with randomly re-initialization weights. The results are summarized below:
>
> **IMP-Refill**
>
> |         IMP Round          |   1   |   2   |   3   |   4   |   5   |
> | :------------------------: | :---: | :---: | :---: | :---: | :---: |
> |    Remaining Weight (%)    | 80.29 | 64.49 | 51.43 | 41.24 | 32.97 |
> |    Ours     | 94.14 | 94.24 | 94.45 | 94.16 | 93.86 |
> | Randomly Re-initialization | 94.04 | 93.96 | 94.20 | 93.98 | 93.53 |
>
>
> **IMP-Regroup**
>
> |         IMP Round          |   1   |   2   |   3   |   4   |   5   |
> | :------------------------: | :---: | :---: | :---: | :---: | :---: |
> |    Remaining Weight (%)    | 80.00 | 72.95 | 69.37 | 67.12 | 58.54 |
> |    Ours     | 94.48 | 94.75 | 94.29 | 94.62 | 94.32 |
> | Randomly Re-initialization | 94.19 | 93.42 | 93.58 | 93.84 | 93.76 |
>
>
> From the table above we can see that with random re-initialization the performances are considerably dropped. This validates that the refilled and regrouped models found are highly non-trivial structural winning tickets.
>
> Although the canonical definition of winning tickets in Jonathan et.al. [r3] is not associated with random re-initialization (it only requires training sparse models from the same random initialization to match the unpruned models’ performance), we agree with reviewer YWUh that it is a good sanity check experiment and we plan to include all the above new results and discussions in our final version.

---

> ### Author Response · Authors · 2021-11-19
> **Response to Reviewer YWUh (Part 2)**
>
> **[Cons 2: Weak baselines?]**
> We respectfully yet strongly disagree with your opinion that “we have a lower baseline accuracy which is easy to match”. We the authors are experienced with LTH and familiar with all benchmark numbers you mentioned. In this paper, we strictly follow [r1] and report the consistently matched performance. Most of your concerns about the baseline accuracy lie in the confusions of our settings such as the number of training epochs and adopted architecture, which we have clarified below. All these experiments have been included in the appendix.
>
> **[Tiny ImageNet]**
> On Tiny ImageNet, the first convolution layer in our ResNet-50 has a kernel size of 7, a padding of 3, and a stride of 2. This is the model structure adopted in [r1,r2], as well as ours. The accuracy of such implementation is about 52%. However, in [r2*] the first convolution layer in ResNet-50 has the kernel size of 3, a padding of 1, and a stride of 1. The accuracy differences lie in the different architecture rather than our settings. To further convince reviewer YWUh, we also conduct extra experiments with the second implementation which has higher testing accuracy. The results are reported below. The dense network’s performance is 65.33% (we train 160 epochs rather than 240 epochs in [r2*] due to the limited time in the rebuttal period).
>
> |         Round          |   1   |   2   |   3   |   4   |
> | :------------------------: | :---: | :---: | :---: | :---: |
> |    IMP - Remaining Weight (%)    | 80.00 | 64.00 | 51.20 | 40.96 |
> |    IMP - Test Accuracy    | 65.44 | 65.69 | 65.50 | 65.73 | 65.23 ｜
> |    Refill - Remaining Weight (%)    | 80.30 | 64.16 | 51.42 | 41.08 | 32.85 |
> |    Refill - Test Accuracy    | 65.27 | 63.40 | 61.89 | 60.43 | 59.64 |
> |  Regroup - Remaining  Weight  | 80.15 | 68.25 | 58.19 | 54.19 | 51.75 |
> | Regroup - Test Accuracy | 65.51 | 65.16 | 65.21 | 64.42 | 64.52 |
>
>
> From the consistent results, we can see that the conclusions still hold. IMP-Regroup can locate structured winning tickets at the sparsity of 42%, and IMP-Refill can locate structured winning tickets at the sparsity of 19.7%.
>
> **[VGG-16 BN and WideResNet-32-2]**
> The training settings we originally used strictly follow [r1]. The number of training epochs is only 160, which is less than 240 in [2*]. Moreover, we spare 5000 images out of the training set as the validation set, and we use the official validation set as the testing set. So what we report here is the test accuracy while [2*] reported the validation accuracy.
> Thus, the accuracy differences lie in the different training epochs and our baseline results are aligned with previous literature  [r1].
> For VGG-16 BN and WideResNet-32-2 on CIFAR-100, we take your suggestion and use the training setting in [2*] to run experiments again. Specifically, we add the number of training epochs to 240, and decay the learning rate at 150th, 180th and 210th by 0.1. For WideResNet-32-2, we report the validation accuracy on the official validation set.
>
> The baseline performance of VGG-16 BN and WideResNet-32-2 is 73.43% and 75.53%. The IMP and IMP with post-processing results are reported below.
>
> **VGG-16 BN (73.43%)**
>
> |         Round          |   1   |   2   |   3   |   4   |   5   | 6 | 7 | 8 | 9 |
> | :------------------------: | :---: | :---: | :---: | :---: | :---: | :---: | :---: | :---: | :---: |
> |    IMP - Remaining Weight (%)    | 80.00 | 64.00 | 51.20 | 40.96 | 32.77 | 26.21 | 20.97 | 16.78 | 13.42 |
> |    IMP - Test Accuracy    | 73.64 | 73.80 | 73.67 | 74.01 | 74.27 | 74.56 | 74.58 | 74.52 | 74.42 |
> |    Refill - Remaining Weight (%)    | 80.17 | 64.06 | 51.31 | 41.08 | 32.85 | 26.33 | 21.03 | 16.94 | 13.42 |
> |    Refill - Test Accuracy    | 73.43 | 72.87 | 72.67 | 71.37 | 70.79 | 71.07 | 69.42 | 68.75 | 67.25 |
> |  Regroup - Remaining  Weight  | 82.36 | 80.00 | 69.46 | 62.61 | 56.09 | 46.53 | 38.18 | 30.98 | 25.27 |
> | Regroup - Test Accuracy | 73.63 | 73.81 | 74.31 | 73.84 | 75.05 | 74.98 | 75.24 | 74.68 | 75.25 |

---

> ### Author Response · Authors · 2021-11-19
> **Response to Reviewer YWUh (Part 2, continued)**
>
> **WideResNet-32-2 (75.53%)**
>
> |         Round          |   1   |   2   |   3   |   4   |   5   | 6 | 7 | 8 | 9 |
> | :------------------------: | :---: | :---: | :---: | :---: | :---: | :---: | :---: | :---: | :---: |
> |    IMP - Remaining Weight (%)    | 80.00 | 64.00 | 51.20 | 40.96 | 32.77 | 26.21 | 20.97 | 16.78 | 13.42 |
> |    IMP - Test Accuracy    | 76.21 | 75.78 | 76.02 | 75.92 | 76.07 | 75.74 | 75.92 | 75.87 | 75.41 |
> |    Refill - Remaining Weight (%)    | 80.00 | 64.06 | 51.51 | 41.51 | 32.84 | 26.53 | 21.11 | 17.11 | 13.67 |
> |    Refill - Test Accuracy    | 75.46 | 74.59 | 73.53 | 72.95 | 72.12 | 70.91 | 69.55 | 67.74 | 65.73 |
> |  Regroup - Remaining  Weight  | 80.00 | 64.00 | 51.20 | 40.96 | 32.99 | 26.27 | 21.76 | 18.14 | 14.85 |
> | Regroup - Test Accuracy | 75.98 | 76.19 | 76.13 | 75.88 | 75.98 | 75.78 | 74.74 | 73.85 | 72.99 |
>
> From the consistent results above, we can see that the conclusions still hold. For example, on WideResNet-32, IMP-Regroup can locate structured winning tickets at the sparsity of 67%, and IMP-Refill can locate structured winning tickets at the sparsity of 20%.
>
> **[ImageNet]**
>
> For ImageNet, the accuracy of the dense ResNet-50 is actually 75.9 but not 75.2. The interval between ticks in Figure 3 is 1%, not 0.2%. Therefore, the gap is around 0.2%. The negligible 0.1%-0.2% gap may be due to random seeds or the different computing facilities.
>
> As suggested by reviewer YWUh, the new results also support that the consistent advantage of our sparse models is generalizable across different training settings and network architectures. Based on these extensive results, we believe it is not “strange” that the accuracy of pruned models can sometimes surpass their dense counterparts at a large sparsity. Also as pointed out by Reviewer TFc4 that the sparsities of unstructured winning tickets are often even greater than 95%.
>
> [r1] Sanity Checks for Lottery Tickets: Does Your Winning Ticket Really Win the Jackpot, NeurIPS 2021
>
> [r2] Progressive Skeletonization: Trimming more fat from a network at initialization, ICLR 2021
>
> [r3] The Lottery Ticket Hypothesis: Finding Sparse, Trainable Neural Networks, ICLR 2019
>
> [1*] https://openaccess.thecvf.com/content_CVPR_2020/papers/Yuan_Revisiting_Knowledge_Distillation_via_Label_Smoothing_Regularization_CVPR_2020_paper.pdf
>
> [2*] https://arxiv.org/pdf/1910.10699.pdf, ICLR 2020

---

> ### Author Response · Authors · 2021-11-19
> **Response to Reviewer YWUh (Part 3)**
>
> **[Cons 3: Customized implementation for structured sparsity? Significance of our proposals?]**
>
> In pruning literature, the structured sparsity does not exclusively mean channel sparsity, but includes channel kernel, and intra-kernel sparsity as well as stated in [r4]. The fine-grained sparse structure is also recently explored in [r5] as pointed out by Reviewer iox5. Therefore, we respectfully disagree with your assertion and believe our work follows the mainstream of structured sparsity.
>
> The practical speedup of regrouping actually comes from shrank kernel size. After regrouping, the dense blocks are extracted out and calculated using general matrix multiplication (GEMM). The implementation only relies on the standard CUDA library so it can perfectly work with common GPUs. Therefore, we do not need special or dedicated accelerators to get practical speedups. It is also a bonus that the regrouping method has no additional requirement, giving great flexibility to the pruning method used.
>
> Moreover, with the fine-grained structured sparsity, we are able to locate winning tickets which have the potentials to be accelerated. It is the first step taken towards the structured lottery ticket hypothesis, pushing forward the frontier of the lottery ticket hypothesis. Reviewer 7uex also acknowledges that “the idea of finding structural winning tickets is meaningful, and the existence of structural winning tickets fills in a crucial piece of the puzzle for the LTH”, and Reviewer iox5 acknowledges that our methods are “simple and intuitive, yet novel and convincing”. Therefore, we believe our work is valuable and interesting, and it is also acknowledged by other reviewers.
>
> [r4] ​​Structured Pruning of Deep Convolutional Neural Networks
>
> [r5] LEARNING N:M FINE-GRAINED STRUCTURED SPARSE NEURAL NETWORKS FROM SCRATCH
>
> **[Cons 4: Effectiveness of refilling on ImageNet]**
>
> In our claim, both channel-wise sparsity and fine-grained sparsity belong to structural sparsity which brings hardware speedup benefits. On ImageNet, as we discussed in the paper, fine-grained structural tickets can be found, while channel-wise structural sparsity is too aggressive to win. Note that scaling up winning ticket findings to large-scale datasets and architectures is actually very challenging. Even for the original Lottery Ticket Hypothesis [JF] with structured sparsity does not hold on ImageNet, so the “Late Rewinding” technique relaxes the condition to find the winning tickets, by rewinding to the early trained weights rather than initialization.
>
> In our context, we adopt block-wise structural sparsity rather than channel-wise sparsity to scale the structural LTH to large datasets and models. We think it is truly an advantage rather than a limitation since our fine-grained structural winning tickets enjoy undamaged trainability, matched performance to dense model, and even more hardware speedups than the channel-wise pruned model with the same sparsity level.
>
> **[Cons 5: Post-training techniques]**
>
> Actually, we regard the independence or “plug-or-play” of our techniques as an advantage rather than a disadvantage. We intentionally do not want to invent another pruning algorithm, but to instead only modify on the basis of unstructured IMP which remains to dominate LTH. We believe such a “control variable” approach is the best to help us understand what is essential to LTH. We love “elegance” too, and the above rationale represents what we feel (subjectively) as scientific elegance.
>
> While traditionally unstructured and structured pruning are largely treated as two independent streams, our methods show they can be bridged to achieve more flexible and delicate accuracy-resource trade-offs. As promising potential benefits beyond LTH, this “independence” can lend our postprocessing techniques to be potentially applicable to other unstructured pruning or sparse training techniques too, such as SNIP/Grasp/RigL, which we’ll explore in future work.
>
> **[Cons 6: Radar plots]**
>
> Thank you for the suggestion. We have included a new set of visualizations via histograms in Figure A11. If you have any other recommended way to display the results, please kindly let us know. We are happy to follow your advice and enrich the visualization of our results.

---

> ### Author Response · Authors · 2021-11-23
> **Sincerely expecting further discussions from Reviewer YWUh**
>
> Dear Reviewer YWUh,
>
> We thank the reviewer YWUh’s time for the review, and we really hope to have a further discussion with reviewer YWUh to see if our response solves the concerns.
>
> We would sincerely appreciate it if reviewer YWUh could reply to the most important points in our rebuttal. For example:
>
> - **[Baseline Accuracy]** The concerns about the baseline accuracy mainly lie in the confusions of our settings such as the number of training epochs and adopted architecture, which is also supported by existing literature and extensive extra experimental results.
>
> - **[Structural Sparsity and Customized Implementation?]** As strongly acknowledged by Reviewer TFc4, Reviewer TFc4 comments that “structured sparsity does not exclusively mean channel sparsity, and the achieved acceleration by this paper is in general meaningful (not tied to particular hardware)”. Our implementation only relies on the standard CUDA library so it can perfectly work with common GPUs.
>
> We genuinely hope reviewer YWUh could kindly check our response. Thank you!
>
> Best wishes,
>
> Authors

---

> ### Author Response · Authors · 2021-11-26
> **Sincerely expecting further discussions from Reviewer YWUh**
>
> Dear Reviewer **YWUh**,
>
> We sincerely hope to have further discussion with reviewer **YWUh** to see if our response solves his/her concerns, since the discussion period is ending in three days.
>
> We are confident that our response should have cleared the air, and we can clarify more if there is more need. We are happy to answer any additional questions and provide more information.
>
> We genuinely hope reviewer **YWUh** could kindly check our response. Thank you!
>
> Best wishes,
>
> Authors

---

> > ### Comment · Reviewer_YWUh · 2021-11-27
> > **Thanks for the response**
> >
> > Thanks for the authors’ feedback! Many of my concerns are well-addressed. Here are the further concerns regarding the feedback.
> >
> > 1.	I understand now why the authors have a significantly lower accuracy with ResNet50 on Tiny ImageNet than the number reported by other works. Really appreciate the explanation!
> >
> > Then compare the new results of ResNet50 on Tiny ImageNet above with Fig. 3 in the paper. In the paper, Fig. 3, when the unpruned model has 53.2% accuracy, Refill can maintain full accuracy at sparsity around 60%; Regroup can maintain full accuracy at sparsity around 75%. Yet now, with a higher baseline accuracy 65.33%, Refill can only maintain full accuracy at sparsity 19.7%; Regroup can only maintain full accuracy at sparsity 42%.
> >
> > Similarly, for VGG-16 BN on CIFAR-100, in the paper (Fig. 5), Refill can maintain full accuracy at sparsity 40%, while now it can only do so at sparsity 20%. For WRN-32-2 on CIFAR-100, in the paper (Fig. 4), Refill can roughly maintain full accuracy at sparsity 40%, yet now it can only do so at sparsity 20%.
> >
> > These are exactly what I said, when the baseline accuracy is low, it is easier to match full accuracy at high sparsity. The authors are fully legitimate to disagree with my concern, but what they presented seems to straightly contradict what they claimed: We respectfully yet strongly disagree with your opinion that “we have a lower baseline accuracy which is easy to match”
> >
> > 2.	“In pruning literature, the structured sparsity does not exclusively mean channel sparsity, but includes channel kernel, and intra-kernel sparsity as well as stated in [r4]. The fine-grained sparse structure is also recently explored in [r5] as pointed out by Reviewer iox5. Therefore, we respectfully disagree with your assertion and believe our work follows the mainstream of structured sparsity.”
> >
> > I fully agree with the point “the structured sparsity does not exclusively mean channel sparsity”, and also recognize their contribution of moving LTH from unstructured sparsity to a more structural sparsity. However, please note, it is not the “structured sparsity” that researchers typically refer to. When researchers mention structured sparsity, the most representative one is filter/channel sparsity, if I am correct. The authors may look at the recent network pruning papers on top-tier conferences (CVPR/ICCV/ECCV/ICML/NeurIPS/ICLR, a good collection is here: https://github.com/he-y/Awesome-Pruning). Most papers on structured pruning are actually doing filter/channel sparsity, not the sparsity like Regroup in this paper. Therefore, the authors claim their paper “follows the mainstream of structured sparsity”. I believe this “mainstream” claim is barely true.
> >
> > This is also what I mentioned, dealing with a self-defined structural sparsity (not the general channel/filter sparsity) making this paper much less interesting (also less useful). It is perfectly fine to only deal with the sparsity like Regroup in this paper, but I guess the title should be corrected to sth like “LTH Can also Have More Coarse-grained Sparsity”. Right now, it seems this paper is over-claiming.
> >
> > I raise my score to 5 given the good feedback from the authors in general. Meanwhile, I’d like to hear the author’s response regarding my further concerns above and see if I need to adjust the score.

---

> > > ### Author Response · Authors · 2021-11-29
> > > **Response to Reviewer YWUh [Extra Cons 1 & 2] (Many Thanks for Raising the Score!)**
> > >
> > > Many thanks reviewer **YWUh** for further comments and suggestions. We are glad to see our response has addressed many of your concerns raised previously. Here are more responses for your further concerns.
> > >
> > > **[Extra Cons 1: Lower baseline accuracy, easy to match?]**
> > >
> > > Thank you so much and we understand your points. We just hope to respectfully point out that the presented results do not contradict the claims. As we previously mentioned in **[Cons 2]**, the different baseline accuracies and matched sparsity levels mainly result from different adopted architectures (e.g., for ResNet-50 cases), different data split (e.g., for WideResNet-32-2), or different training epochs (e.g., for VGG-16). Then, it is inappropriate to derive the causal relationship between “lower baseline accuracy” and “higher matched sparsity”, since various influential factors are not strictly controlled.
> > >
> > > We believe that there might exist many underlying reasons why the matching sparsity changes, such as the backbone architecture and training schemes, while accuracy is a superficial observation depending on those factors. A possible way to study the relationship is by building a causal probabilistic graph as shown in [1], which we will explore more in our future works. Many thanks for raising this interesting scientific question.
> > >
> > > [1] Fantastic Generalization Measures and Where to Find Them
> > >
> > > **[Extra Cons 2: Tune down the claim of structural sparsity]**
> > >
> > > We truly thank reviewer **YWUh**’s feedback with a comprehensive list of pruning works. We agree that a large portion of existing literature on structured pruning focuses on channel/kernel sparsity. Meanwhile, we want to respectfully point out that fine-grained structural sparsity has also become prevailing recently, sparking great interest in industries like NVIDIA [2] and Google [5].
> > >
> > > Moreover, our paper investigates both channel-wise structural sparsity (i.e., Refill(+)) and fine-grained group-wise structural sparsity (i.e., Regroup). Although only Regroup works effectively in identifying winning tickets on ImageNet, for most network architectures and datasets, both Refill(+) and Regroup can locate winning tickets, as shown in Figure 3/4/5 and our extra experiments. Note that the fine-grained structural sparsity is not newly proposed or defined by us, but rather well-accepted by prior literature [1,2,3,4].
> > >
> > > Lastly, we would like to follow reviewer **YWUh**’s suggestion and change the title of our work as soon as the portal reopens. The new title will emphasize the “Coarse-grained Sparsity”. Our tentative new title is now **“Coarsening the granularity: towards structurally sparse lottery tickets”** - and we welcome your further feedbacks or suggestions!
> > >
> > > [1] Exploring the Granularity of Sparsity in Convolutional Neural Networks
> > >
> > > [2] Exploiting NVIDIA Ampere Structured Sparsity with cuSPARSELt, https://developer.nvidia.com/blog/exploiting-ampere-structured-sparsity-with-cusparselt/
> > >
> > > [3] Learning N: M fine-grained structured sparse neural networks from scratch
> > >
> > > [4] GRIM: A General, Real-Time Deep Learning Inference Framework for Mobile Devices based on Fine-Grained Structured Weight Sparsity
> > >
> > > [5] Optimizing Speech Recognition For The Edge

---

> > > > ### Comment · Reviewer_YWUh · 2021-11-29
> > > > **Thanks for the further responses!**
> > > >
> > > > “We just hope to respectfully point out that the presented results do not contradict the claims” -- This claim is true, yet not very informative. Because in the original LTH paper, it only says the subnet can be trained to full accuracy at *a certain* sparsity. It does not specifically restrict that the sparsity has to be greater than some threshold. Even at sparsity 1%, you can still claim you find the winning tickets.
> > > >
> > > > But clearly, the problem here is that, if the sparsity of matching full accuracy gets lower (such as from 60% to 19.7%, 75% to 42%, shown above, the drop is significant, not marginal), the practical benefit is discounted much and even becomes trivial (such as the 19.7% structural sparsity, which is very marginal and barely useful in practice). Note for the original LTH, they typically can match full accuracy at sparsity like 90%. I understand structured sparsity is harder than unstructured sparsity, so probably you cannot find winning tickets still at like 90%, but what is of interest may be a number like 60-70%. Right now, the matching-accuracy sparsity (using the *higher* baselines, which I think is the correct way to have a really fair comparison) is only 20%~40%, and, it is not the most prevailing filter/channel sparsity, but a fine-grained sparsity. Taking all into consideration, this is why I said this paper does not really achieve what the title claims right now and the title may need a change to match the real contribution.
> > > >
> > > > I fully recognize their contributions in making LTH one more step forward towards more structural sparsity. Meanwhile,  given (1) the real matching-accuracy sparsities are not significant if using the stronger baselines, (2) the sparsity is not the most typical structured sparsity (filter/channel sparsity), I keep the rating at 5.

---

### Official Review · Reviewer_TFc4 · 2021-11-02

**Correctness:** 4
**Technical Novelty And Significance:** 3
**Empirical Novelty And Significance:** 4
**Recommendation:** 8
**Confidence:** 4

**Main Review:**

The authors explored hardware-friendly structural sparsity (including channel-wise and group-wise patterns) to find lottery tickets. They propose new refilling techniques based on the classical iterative unstructured pruning process to create channel-wise structural sparsity by growing back the pruned elements within the most important channels and abandoning the rest.

On top of that, they introduce another regrouping algorithm based on hypergraph partitioning (Rumi et al., 2020) to establish group-wise structural patterns which are more amenable to pruning due to the shape flexibility of grouped blocks.

The authors reported many experiments on CIFAR-10, CIFAR-100, TinyImageNet, and ImageNet, using various backbones like ResNets, VGGs, and MobileNets. Overall, the structural winning tickets are shown to achieve 53.75% ∼ 64.93% GPU running time savings at 45% ∼ 80% channel- or group-wise sparsity.

One should note that the sparsity level discussed in this work is not as the usual high as in unstructured tickets (often >95%), but the hardware acceleration is way more apparent. The ablation studies and visualization are thorough.

I have just a few clarifications or curiosity questions:
- It seems the acceleration effect on the structural ticket (especially after regrouping) requires some specific hardware accelerator and efficient implementations such as GEMM. It is then unclear whether the comparison with vanilla LTH is fair or not: could unstructured tickets also benefit from those? How much performance gain of the reported numbers comes from hardware specialization (Rumi 2020)?
- Whether the same techniques can be applied to lottery tickets found not from random initialization, e.g., how about the pre-trained model lottery tickets as in (Chen et al., 2020b)? Will the refilling/regrouping rely on random weight initialization anyhow?
- How much is the extra overhead of refilling/regrouping compared to the IMP cost?

**Summary Of The Paper:**

The paper presents an exciting new finding – by properly reorganizing elements during the IMP process, it’s possible to achieve structurally sparse winning tickets at high sparsity levels that can be easily hardware accelerated.

**Summary Of The Review:**

The paper is in a good shape that displays the first set of positive results bridging the gap between the lottery ticket hypothesis and practical accelerations on real-world hardware. Algorithm details, as well as results, are sufficient and convincing.

---

> ### Author Response · Authors · 2021-11-19
> **Response to Reviewer TFc4**
>
> Thank you for praising the value of work and that our positive results can bridge the gap between LTH and accelerations on real-world hardware. We have addressed your concerns below.
>
> **[Question 1: Do the code implementations and hardware help acceleration?]**
>
> We respectfully argue that our methods do not need any specific hardware accelerators. Our methods are examined on general GPUs. GEMM, which is short for General Matrix Multiplication, is also not a special convolution algorithm.
>
> We assure you that our comparison with the vanilla LTH is fair. We evaluate the model under the same training/testing protocols, and profile their performance using the same devices and libraries. However, unstructured pruning will lead to “irregularity in memory access and computation, creating huge throughput gap, making it a major challenge to improve the performance of CNN inference” (Rumi 2020). So they are hard to be speedup-ed, and can only have the same inference time as the dense model with the general implementation.
>
> We use general hardware in our experiment. We follow (Rumi 2020) to use 2080 Ti to calculate the inference time and the speedup ratio. Since we use the general-purpose GPUs, the acceleration does not really hinge on specialized hardware such as dedicated accelerators, and hence the benefits are broadly applicable. We will make this par clear in revision.
>
> **[Question 2: Can the method be applied to other tickets? ]**
>
> Thank you for your suggestion. It can certainly be applied to other forms of tickets.
>
> We actually have already proved that our method does not rely on random weight initialization in Figure 7 (middle). We can see that the performance of group-wise structural tickets with different weight rewinding clearly surpass random reinitializing at sparsity levels > 30% - their validity does not depend on the random initializations.
>
> We also follow your suggestion and try the supervised ticket described in [r1]. We first find tickets in ResNet-50 on pretraining tasks (ImageNet), and then re-train on CIFAR-10. The dense performance is 95.49%. We can see from the table below that refill and regroup can also work well with pre-training tickets. Specifically, structural winning tickets with 56.01% and 35.86% sparsity can be located by Regroup and Refll while the test accuracy is 95.51% and 95.81%, which means that our methods can work with pre-trained tickets, and do not rely on the random init.
>
> | Regroup - Remaining Weight | Regroup - Test Accuracy | Refill - Remaining Weight | Refill - Test Accuracy |
> | :---: | :---: | :---: | :---: |
> | 77.28% | 94.94 | 64.14% | 95.81 |
> | 71.40% | 96.28 | 51.37% | 95.14 |
> | 67.60% | 94.09 | 41.01% | 94.51 |
> | 59.43% | 95.65 | 32.76% | 94.38 |
> | 51.84% | 95.39 | 26.17% | 94.19 |
> | 43.99% | 95.51 | 20.97% | 94.11 |
>
> [r1] The lottery tickets hypothesis for supervised and self-supervised pre-training in computer vision models. CVPR2021
>
> **[Question 3: Overhead of refilling and regrouping]**
>
> There is little overhead of refilling since we only need to calculate the layerwise sparsity of each layer and then rearrange the sparse mask. More specifically, the time spent on refilling a ResNet-18 is merely 0.5 seconds.
>
> The overhead of regrouping is mainly due to the partitioning process. The time on partitioning a 512x4608 (512x512x3x3) graph is about 3.452 seconds.
>
> In comparison, the time for training ResNet-18 on CIFAR-10 for an epoch is 21.41 seconds, which means roughly 57 minutes for 160 epochs, and 570 minutes for 10 IMP rounds.
>
> In conclusion, the two methods have little overhead compared to IMP cost.

---

### Official Review · Reviewer_7uex · 2021-11-08

**Correctness:** 4
**Technical Novelty And Significance:** 3
**Empirical Novelty And Significance:** 3
**Recommendation:** 6
**Confidence:** 4

**Details Of Ethics Concerns:**

Nothing.

**Main Review:**

# Strength:

The structural sparsity and unstructured sparsity are not new. However, in previous works about the lottery ticket hypothesis (LTH), the winning tickets are typically unstructured. Therefore, the idea of finding structural winning tickets is meaningful, and the existence of structural winning tickets fills in a crucial piece of the puzzle for the LTH. In this work, the authors propose a method to find structural winning tickets. Extensive experiments on various datasets and network structures show the method is effective and efficient.

# Weakness:

1. In some experiments (such as Figs. 3 and 5), the authors report time-saving when using a specific GPU and batch size. The time saving might vary when other GPUs or batch sizes are used. Therefore, I believe it would be better to also include the number of FLOPs in the results.
2. IMP-Refill and IMP-Refill+ generate different results. Why only IMP-Refill is included in Fig. 8?
3. In Figs. 3, 4, and 5, it seems different methods stop at different percentages of sparsity. Why don't they all prune to the same percentage?

**Summary Of The Paper:**

This work proposes a method to effectively find structurally sparse winning tickets. It consists of some post-processing techniques that can be added to each round of standard iterative magnitude pruning (IMP) methods. Starting from unstructured sparse sub-networks, the method uses a "re-filling and re-grouping" manner to enforce the formation of structural sparsity.

**Summary Of The Review:**

This work demonstrate that lottery tickets can not only be unstructured but also be structural, which makes it a novel and interesting work. There are minor issues in experiments.

---

> ### Author Response · Authors · 2021-11-19
> **Response to Reviewer 7uex**
>
> Thank you for acknowledging the effectiveness of our refilling and regrouping techniques. We find your suggestions are insightful and valuable for us to further improve our work.
>
> **[Cons 1: FLOPs of models]**
>
> Thank you for your suggestion. We present the calculated FLOPs of VGG-16 on CIFAR-10 with different methods and sparsity below. We also promise to include FLOPs for all models in the final version.
>
> |   Method    | Remaining Weight | FLOPs |
> | :---------: | :------: | :---: |
> |     IMP     |     100%     |    0.314G   |
> | IMP-Refill  |      32.84%    |  0.089G    |
> | IMP-Refill+ |     46.41%    |    0.122G   |
> | IMP-Regroup |   30.12%      |   0.136G    |
>
> It is noteworthy that Refill and Refill+ will lower the input and the output channel of a convolution layer while Regroup cannot, so Refill and Refill+ can save more FLOPs under the same level of sparsity.
>
> We have conducted experiments to study whether the different batch sizes and different GPUs can change the speedup performance. On 1080TI with batch size 64, the overall speedup is 54.28%. When the batch size is 128 the speedup is 53.88% and 54.86% when the batch size is 256. On 2080TI with batch size 64, the speedup is 57.75%. We can see that, neither the batch size nor specific GPU will significantly change the speedup ratio.
>
> **[Cons 2: No refill+ in Figure 8]**
>
> Thank you for pointing it out. We take your suggestion and put the patterns generated by Refill+ in Figure A10. However, we also want to clarify that since the Refill and Refill+ can both generate channel-wise structured sparsity, these two channel-wise sparsities share highly similar patterns. Therefore, we did not put both of them in the original version due to the space limitation.
>
> **[Cons 3: Stopping sparsity]**
>
> Thanks. We want to clarify that we investigate 10 unstructured pruned checkpoints. Refill, Refll+, and Regroup are applied to this same set of unstructured pruned models. These methods recover a different proportion of pruned parameters from the sparse models. So after Refill, Refill+, and Regroup, the sparsity is different for a given unstructured pruned model. That is why the stopping sparsities seem to be different in our case.

---

> > ### Comment · Reviewer_7uex · 2021-12-01
> > **Post rebuttal responses**
> >
> > This work is novel and constructive and has only minor issues. After reading the responses from the authors, I find they solve my concerns. Therefore, I would recommend to accept this paper.

---

### Decision · Program_Chairs · 2022-01-20

**Decision:**

Reject

**Comment:**

### Summary

This work demonstrates that it is possible to identify lottery tickets with some manner of structural sparsity. The work finds success through refilling (perhaps better termed infilling) and regrouping, two techniques that have found previous homes in other parts of the literature.

### Discussion

#### Strengths

- Tackles an interesting problem.

- At face value, I believe the paper achieves its claimed goal, though not with full clarity as written.

#### Weaknesses

- There is room to clarify the atypical form of structure here for readers. The suggested title change would appropriately set expectations. However, refinements in the text as well would be welcomed.

- As I discuss below, the claims should be settled with respect to the strongest baselines.

- Random reinitialization should play a primary role in the presentation of the text. As the authors note, the original lottery ticket paper did not require besting the performance of random reinitialization. However, the random reinitialization results are central to the main figures of the original paper and demonstrate that the result is not merely happenstance. This paper should follow that practice.

### Recommendation

I recommend Reject and I do not do so lightly, given the scores. The work here is promising because finding a path to better-performing lottery tickets remains an open challenge. However, Reviewer YWUh has voiced reasonable concerns about the evaluation methodology.

I've read the detailed authors' responses and agree with the authors that the presented results may depend on choices in the hyperparameters and training strategy. Having said that, it is critically important for the paper to include in the primary text the strongest baselines available, despite this dependence. On such baselines, the results are worse than originally reported and hence, the primary claims must either be revised to be these results or revised to include these results, with the primary claims providing a range of results. Though the authors offer to make revisions for the camera-ready, the required revisions here are substantial enough to require additional review, which is out of the scope of the current process.

An additional oversight in this methodology -- at least as reported -- is that the primary target of the lottery ticket hypothesis is sparse training, not sparse inference. Evaluating solely the inference performance -- rather than training performance, which includes both the forward and backward pass -- is, therefore, inconsistent with the purpose of lottery tickets. This methodological error will too need to be repaired in the final version of this work.